# Pain, Anger, and the Fear of Being Discovered Persist Long after the Disclosure of HIV Serostatus among Adolescents with Perinatal HIV in Rural Communities in South Africa

**DOI:** 10.3390/children7120261

**Published:** 2020-11-28

**Authors:** Yvonne Maseko, Sphiwe Madiba

**Affiliations:** Department of Public Health, School of Health Care Sciences, Sefako Makgatho Health Sciences University, Pretoria 0001, South Africa; hlomo2904@gmail.com

**Keywords:** delayed disclosure, experiences, impact, long term ART, negative emotions, wellbeing

## Abstract

Informing adolescents of their HIV serostatus forms part of their HIV care and is a critical step in the transition to adult clinical care services. This article describes the experiences of adolescents with perinatal HIV in regard to disclosure, and examines the impact disclosure has on their emotional health and behaviors. We used a qualitative design to conduct interviews with 21 adolescents aged 12–19 years recruited from a rural district in South Africa. NVivo 10 computer software was used for thematic analyses. All adolescents were aware of their HIV-serostatus. The findings show that delayed disclosure, was a one-time event, and was unplanned. Disclosure occurred at the clinic rather than the adolescent’s home. For most adolescents, feelings of anger, pain, sadness, negative perceptions of self, internalized stigma, and denial persisted long after disclosure occurred. They lived in constant fear of having their serostatus being discovered, and they developed a sense of fear of self-disclosure. Their negative emotions undermined treatment adherence. In contrast, other adolescents that described disclosure as a positive event, had accepted their HIV status, and lived similar to other adolescents. The prolonged negative reactions underscore the importance of ongoing post-disclosure interventions for adolescents in rural settings where psychosocial support services are insufficient to address their emotional wellbeing.

## 1. Introduction

Informing children and adolescents of their HIV serostatus is critical and forms part of the comprehensive HIV care received. As the number of adolescents with perinatal HIV (PHIV) increases, due to the increased access to antiretroviral therapy (ART) [1,2], disclosure is an important step towards the long-term disease management necessary to the transition between pediatric and adult clinical care services [3,4]. The maturing HIV epidemic and the increased accessibility of ART have led to children with PHIV reaching adolescence in large numbers [5]. In 2016, around two million adolescents aged 12–19 years were living with HIV, and nearly 85% of them live in sub-Saharan Africa (SSA) [6]. When such adolescents reach puberty and become sexually active, they carry the risk of transmitting HIV to others. Thus, knowing about their HIV serostatus is an important part of HIV prevention that could reduce high-risk sexual behaviors among adolescents.

In high HIV-prevalence settings in SSA, a high proportion of children and adolescents with PHIV are unaware of their diagnosis, despite their attending clinics regularly and taking lifelong ART [1,2,3]. Disclosure is a complex and difficult undertaking when it involves adolescents with PHIV [4,5,6] because disclosure may have either negative or positive consequences [7,8,9]. Nevertheless, the medical and psychosocial benefits of disclosure have been well documented in developed and developing countries. Disclosure has been associated with adherence to ART, better engagement with HIV care, better HIV treatment outcomes, higher self-esteem, and improved mental health outcomes [5,10,11,12].

Disclosure is complicated because caregivers choose not to disclose to children with PHIV of their serostatus, due to a range of factors [13,14,15,16]. Many child and caregiver factors influence why caregivers delay disclosure. Child-related factors include being afraid of negative psychological effects for the child, concerns about the inadequate maturity of adolescents, and concerns that children may accidentally disclose their status to others. Caregiver-related factors include the fear of HIV-related stigma, fear of being blamed by the children, lack of disclosure skill, and guilt regarding the HIV transmission [7,15,17,18,19,20]. Prolonged silence and secrecy about an adolescent’s HIV status could increase the risk of psychosocial distress leading to poor disease progression and drug resistance [21,22].

Adolescents, like adults, struggle to self-disclose to significant others and perceive disclosure as a complex and difficult process with concerns and fears about the outcomes of disclosure. They consider information about their HIV serostatus to be private [23], and prefer to reveal their status to others on their own terms [5,24]. They are often reluctant to disclose to sexual partners, mainly for fear of rejection, stigmatization, and abandonment [4,7,25,26]. By keeping their HIV status a secret, they ensure the continuation of a positive relationship with their peers [7].

Although research has shown the positive impacts of disclosure on the emotional health of children with PHIV, evidence suggests that disclosure outcomes among adolescents vary in relation to their social context [27], with many adolescents struggling with the stresses that accompany disclosure [3]. Negative reactions and emotions, such as fear, confusion, disbelief, worry, anger, pain, and sadness, immediately the following disclosure have been documented [3,5,28,29]. Data show that disclosing serostatus to adolescents could also generate complex negative reactions, including anxiety, distress, withdrawal, and depression [19,20,26,30]. Disclosure may also impact the sexual relationships of adolescents and decrease the desire to adhere to medication [7].

The literature suggests that disclosure does not seem to have major long-term negative psychological outcomes, but may cause short-term anxiety, which is mitigated over time [3,8,14]. However, Mburu et al. [7] argue that most studies do not have evidence that some of the negative psychological outcomes of disclosure could improve over time, since most are cross-sectional studies that provide only a snapshot of adolescents’ experiences at one point in time. In addition, most studies have focused on participants in early adolescence (10 to14 years of age), but as the adolescents grow older, there is a greater self-awareness and an exaggeration of the need to be accepted by peers [19]. Older adolescents are also most likely to have concerns about the impact of their HIV serostatus on relationships with their sexual partners and peers [31].

Few studies in SSA document the experiences of adolescents following the disclosure of their diagnosis, as well as the impact of disclosure on their psychosocial health [9,19,26,32]. Even fewer studies have described the post-disclosure psychosocial needs of older adolescents who experience delayed disclosure [29,33]. South Africa has one of the largest populations of children and adolescents on ART program in the world, estimated at 730,882 [34]. While disclosure to children and adolescents forms part of the comprehensive HIV care received, there is lack of data on how disclosure practices and processes are experienced by adolescents [5], and in particular, those in rural settings, given the different social contexts of urban and rural families and the different health facilities. In addition, psychosocial interventions, such as support groups, psychosocial Teen Clubs, and youth-friendly HIV clinics for adolescents with PHIV are run through non-governmental organizations concentrated in urban settings.

In this paper, we describe the experiences of adolescents with PHIV in regard to disclosure, specifically examining the impact disclosure has on the adolescent’s emotional health and behaviors. It is crucial to investigate the impact of disclosure on the emotional and mental health outcomes of adolescents in rural settings to inform psychosocial interventions.

## 2. Materials and Methods

### 2.1. Design of the Study

This was a qualitative design using in-depth interviews to collect data from adolescents with PHIV. The setting of the study was two primary health care facilities located in a rural health district in Mpumalanga Province, South Africa. The facilities provide ART and other clinical services, including HIV testing and counseling, ART initiation, clinical consultations, the prescription of ART, and the screening and treatment of opportunistic infections in children, adolescents, and adults living with HIV. Facilities in the health district have been offering HIV and AIDS services to children and adolescents through Nurse Initiated Management of Antiretroviral Therapy (NIMART) since 2009. An estimated 1500 adolescents with PHIV accessed their long-term ART from such health facilities through the NIMART services. The two facilities were selected because they have a high volume of adolescents with PHIV on ART.

### 2.2. Sampling

The recruitment of study participants was done via purposive sampling facilitated by the lead author with the assistance of a trained research assistant (the researchers). Pre-meeting arrangements were made by the researchers with the facility managers to provide an overview of the study’s objectives. The adolescents were sampled if they were perinatally infected, were aged 12–19 years old, and were aware of their HIV status. Those who met the criteria for inclusion in the study were recruited through the assistance of the clinic staff. They were approached in the mornings during the clinics’ visiting days when they arrived for their ART and or for a routine check-up. The researchers confirmed with the caregivers and the clinic staff that the adolescents were aware of their HIV status before inviting them to participate in the study.

When recruiting participants who were minors, the researchers first approached the caregivers to ask for permission to involve their children in the study. They then explained to both the caregivers and the adolescents whose caregivers had given permission the objectives of the study, the nature of informed consent and assent, the provision of confidentiality, and the fact that their participation would be voluntary. The researchers approached the adolescents in the presence of their caregivers, asked for their assent to participate in the study, and notified them that they would be interviewed in the absence of their caregivers. In total, 21 adolescents aged 12–19 were recruited from the selected health facilities. The sample size was guided by data saturation, which was achieved when the interviews no longer generated any new themes and all emerging themes had been explored [35,36].

### 2.3. Data Collection

The researchers used a semi-structured guide to interview the participants. The interviews focused on their accounts of the disclosure process, the ideal age for disclosure, reactions and responses to disclosure, and post-disclosure experiences and behaviors. In addition, probes and follow-up questions were used to clarify or verify the participants’ responses. Prior to the data collection, the researchers received training on interviewing techniques with children and on ethical issues surrounding confidentiality and voluntary participation. The interviews were held in a private interview room at the health facilities after the participant had completed his or her routine medical check-up and received an ART refill. The languages used for the interviews was Xitsonga and SiSwati to allow the participants to speak freely about their experiences. All the interviews were conducted in the absence of the caregivers, after obtaining written informed consent and assent. Each interview lasted for between 20–30 min and was recorded with the permission of the participant.

The participants completed a short demographic tool at the end of the interview so that information could be collected about gender, age, duration of ART, age at the point of disclosure, schooling status, and family structure.

### 2.4. Data Analysis

Following the approach of Braun and Clarke [37], a thematic analysis was performed using an inductive approach to identify codes in the data. The verbatim transcription, translation, and repeated reading of the transcripts were the first steps in the data analysis, allowing the authors to develop familiarity with the data and identify emergent codes. The authors discussed and agreed on the initial codes that were used to develop a codebook. Once the codebook had been completed, the transcripts were uploaded into NVivo 10, which is a qualitative data analysis software [38] for the application of codes. The codes were refined and organized into the themes and sub-themes that had emerged from the data. The themes and sub-themes were reviewed and redefined until the researchers had reached a consensus on the final themes to be used to present the data.

The authors used several strategies to promote rigor and establish the credibility of the study. First, the interviews were performed in local languages, audio recoded, and transcribed verbatim to reflect the true experience of the participants. The authors held debriefing sessions continuously throughout the research process. In addition, both authors analyzed the data independently to minimize the effect of investigator bias [39].

### 2.5. Ethical Considerations

The study was conducted in accordance with the Declaration of Helsinki, and the protocol was approved by the Ethics Committee of Sefako Makgatho Health Sciences University (SMUREC/H/19/2018: PG). The Provincial Health Research Committee and the District Manager gave permission to implement the study in the selected health facilities. Assent, privacy, and confidentiality were addressed in a manner consistent with national and international guidelines for conducting research with minors [40,41]. The parents and guardians gave their informed consent for adolescents below the age of 18 years, while those over 18 years of age and above gave informed consent. Those below the age of 18 years also gave written assent. The researchers explained to the adolescents and caregivers that participation was voluntary and that they could withdraw from the study at any stage, to minimize any pressure to participate that they might feel. Pseudonyms were used during the interviews to protect the identities of adolescents.

## 3. Results

### 3.1. Demographic and Clinical Characteristics of Participants 

The sample consisted of 21 adolescents with PHIV aged between 12 and 19 years. All were aware of their HIV status. Their mean age was 15.9 years; twelve were older than 16 years, and 14 of the 21 were females (Table 1). Four reported being disclosed to by their mothers at home, sixteen reported being disclosed to by a healthcare worker (HCW) at the health care facility, and one reported being disclosed to by a close family member).

The ages when the participants were informed about their HIV status ranged from 10 to 19 years. Nine had been told about their status between 10 and 14 years, while 12 learned about their status between 15 and 19 years (Table 2). Concerning their adherence to ART, 17 participants reported that they were adhering to their medication regime. 

### 3.2. Themes

Table 3 shows a summary of the two main themes and eight sub-themes that emerged from the analysis of the interviews.

#### 3.2.1. Circumstances Leading to Disclosure

Disclosure to adolescents is often postponed to later years and occurs when the illness gets worse. Under these circumstances, disclosure is unplanned and is triggered by poor adherence to ART or the severity of the disease.


*I was sick and my mom brought me to the clinic to get treatment and the nurses advised her to have me tested. That’s when I found out that I was HIV-positive.*
*(Tintswalo, 17 years, tested at age 14)*


*I had bad sores on my hands, and was taken to the clinic where they run the test. They told my mom that I am HIV-positive.*
*(Sonnet, 14 years, tested at age 14)*

Disclosure also occurred when the adolescents started asking about taking medication.


*I asked my mom why I was taking treatment and she told me I was HIV-positive.*
*(Nomfundo, 16 years, tested at age 15)*

Some adolescents were tested for HIV because they already suspected that they were infected with HIV.


*I was sick, as I was the one taking care of my father before he died. I realized that there was something wrong with both my parents, so I decided to take the test. That’s when I found out that I was HIV-positive.*
*(Milano, 19 years, tested at age 16)*

#### 3.2.2. Preferred Ideal Age of Disclosure

The disclosure was delayed for most of the participants, 12 out of 21 were disclosed to between 15 and 19 years of age. The participants felt that it would have been easier to accept their HIV status if disclosure had been done when they were younger.


*It would be better if disclosure were done when I was young. I would have accepted long time ago. Knowing the status when you are older is hard. Sometimes it increases my stress level. I was stressing a lot, and that affected my schoolwork.*
*(Vutivi, 19 years, tested at age 18)*

Those whose disclosure was done when they were younger were comfortable with the timing.


*I think I was told at the right time because I am in a position to know and understand my status and it helped me because I took my medication before I be ill.*
*(Buhle, 13 years, disclosed to at age 12)*


*I think I was told at the right time. I grew up knowing my status. If they disclosed later it was going to be painful to me. Now I am not angry.*
*(Appreciate, 15 years, disclosed to at age 14)*

#### 3.2.3. Reaction and Response to Disclosure

The participants reacted to disclosure with varying degrees of anger against the biological mother, shock, fear, and self-blame.

For most of the participants, receiving the disclosure of their HIV-positive diagnosis was emotionally difficult. They reacted with shock and confusion, due to the unexpected nature of the HIV diagnosis.


*I was shocked and that did not sit well with me, because I never slept with any man. If I slept with a man, I believe it was going to be better because I will know I brought this thing upon me.*
*(Alucia, 19 years, disclosed at age 17)*


*I was very shocked as I did not think I would be taking HIV medication for the rest of my life. I was just taking medication without knowing the medication is for HIV.*
*(Buhle, 13 years, disclosed at age 12)*

The participants described the disclosure event as painful and shocking.


*I am hurting, considering I was born with the virus. It would have been better if I got HIV on my own, not born with it.*
*(Vutivi, 19 years, disclosed at age 18)*


*I am still hurting and the fact that my father passed away while I’m still young make the situation to be more painful because I think If he was around my life was going to be much better.*
*(Nomfundo, 16 years, disclosed at age 15)*

The lack of disclosure from the mothers led to a continuous search for answers.


*I was hurting. I asked myself why my mom never told me that she is HIV-positive. I had to find out from my granny because my mother passed away. I think if she disclosed her status to me I was going to accept my status.*
*(Thandazo, 14 years, tested at age 12)*

Some said that they felt angry upon discovering that they were HIV-positive.


*I was angry with my mother, I wanted to know where I got HIV, and my mother could not tell me where I got HIV.*
*(Vutomi, 16 years, disclosed at age 15)*


*I am angry because I am taking HIV medication and because they did not tell me early. Being HIV positive makes me angry.*
*(Mkhungelo, 14 years, disclosed at age 14)*

A few expressed feelings of fear after the disclosure. The HIV diagnosis resulted in fear because it was unexpected and HIV was perceived to be an incurable and fatal disease.


*I was scared and asked myself why this is happening to me. I saw many people living with HIV and never thought one day I will have it and live with it for the rest of my life.*
*(Nomfundo, 16 years, disclosed at age 15)*


*I don’t like HIV because HIV kills people sometimes, when they do not take their medication correctly.*
*(Mkhungelo, 14 years, disclosed at age 14)*

On the other hand, a few participants did not experience negative emotional reactions after learning about their HIV diagnosis.


*I was relieved as any person can get this disease, so I did not have any problem being HIV. Being angry will not change the fact that I am HIV-positive.*
*(Appreciate, 15 years, disclosed at age 14)*

#### 3.2.4. Self-Disclosure of Status to Others

Most participants had not self-disclosed because they wanted to control when and to whom to disclose their HIV status.


*I will never disclose my status to any friends because they will talk about me and not play with me because they will say I will infect them with my disease.*
*(Vonile, 16 years, disclosed at age 16)*


*I will not disclose. They will stop associating with me and say I will infect them with HIV. Maybe when I am an adult I will disclose to both my friends and girlfriend.*
*(Wilton, 16 years, disclosed at age 15)*

The interviews revealed that there was limited disclosure to peers in school settings, even though the participants spend a lot of time at school.


*I will never disclose to any friends. I do not trust anyone. Even my close relatives do not know about my status. The only person I trusted was my father and now he is gone, so I do not talk to anyone about my problems. Some friends are not what you think they are. If I disclose to one of them they will go around talking about me.*
*(Milano, 19 years, tested at age 16)*

Disclosure to a romantic partner was also difficult; the participants feared that disclosure could end their relationships and that there would be other negative consequences.


*My boyfriend…, I don’t trust him because I’m not sure about his future plans with me and if it happens that we separate he may go around telling other people that I’m HIV-positive. He might also hate me that I am HIV positive.*
*(Buhle, 13 years, tested at age 12)*


*I can tell my boyfriend after some time when I know I can trust him because if we separate, he can also tell other people about my status.*
*(Tintswalo, 17 years, tested at age 14)*

#### 3.2.5. Coping with the HIV Diagnosis

Some of the participants reported that knowledge of their HIV status contributed to enhanced adherence to ART.


*Life is good to me. I am healthy, I am not sick, and I am always happy. There is nothing stressing me, I am aware of my status and I take good care of myself.*
*(Nqobile, 16 years, disclosed at age 15)*


*I was OK because I suspected that I might be HIV-positive since my mom died and she was HIV-positive. I live like any other young children living without HIV, and as long as I take my medication, I will be like them.*
*(Thandazo, 14 years, tested at age 12)*

Some indicated that the knowledge that they were not the only persons living with HIV give them encouragement.


*I am not the only one living with HIV. At the hospital, they told me that many people are HIV-positive and healthy because they are taking the medication. That is why I feel good because I will take my medication and be healthy like any other person.*
*(Sindiswa, 17 years, tested at age 15)*

Some were able to accept their HIV-positive status and interact with friends and peers at school.


*I am not different from other people. The only problem was when I started taking the medication you feel sick but after some time you get used to the medication and life goes on.*
*(Alucia, 19 years, tested at age 17)*

#### 3.2.6. Negative Perceptions of Self

In contrast, participants who did not accept their status found living with HIV unbearable as they continued to ask why it had to happen to them.


*Sometimes at school when other children talk about HIV, I end up thinking they are talking about me. This makes me feel as if I am different from my peers.*
*(Vutivi, 19 years, disclosed at age 18)*


*What pains me the most is that I have never slept with anyone, but I am already HIV-positive.*
*(Tibane, 15 years, disclosed at age 15)*


*HIV is bad. When you are HIV-positive people tend to think you are having sexual intercourse with many people.*
*(Vutomi, 16 years, disclosed at age 15)*

#### 3.2.7. Being the Only HIV-Positive Sibling

The interviews revealed that some of the participants had internalized blame, as they could not understand how they had been infected.


*I do not understand why I am the only one who is HIV-positive while my twin sister is HIV-negative. At the beginning I did not understand what is going to happen with my life, how I got the HIV, because my mother did not tell me about my HIV status.*
*(Buhle, 13 years, tested at age 12)*


*My twin sister does not have the virus and I always ask myself why I am the only one who is infected.*
*(Vutivi, 19 years, tested at age 18)*

#### 3.2.8. Taking ART Interferes with Adolescents’ Lives

Taking ART was a constant reminder to the participants that they are living with HIV. They live in constant fear of being found out to be positive and resorted to having to hide the medication from their friends.


*It is very painful that you will never live your life like any other young people. The problem is that you have to take medication for the rest of your life. Let’s say we go out to have fun when it is time to take medication, I have to lie to my friends so that I can go home to take medication as I can’t carry my medication with me because they might see the pills and they will make fun of me.*
*(Delicious, 14 years, tested at age 13)*


*I can’t enjoy being with my friend because when I am with them I must always check the time so that I can run home to take my medication. That is what stresses me the most and I do not enjoy my life.*
*(Tintswalo, 17 years, tested at age 14)*

The participants described the impact of their frequent visits to the clinics on their personal lives. They felt that visiting the clinic for routine medical consultations or to collect ART interrupted their daily routines.


*It is not a nice feeling going to the clinic. I always go while other children do not. Neighbors and children at school ask why I always go to the clinic. I think of the status almost every day of my life.*
*(Vutivi, 19 years, tested at age 18)*

## 4. Discussion

This study explored the impact of the disclosure of HIV status on the emotions and behaviors of adolescents with PHIV in rural settings. The participants of this study consider that they came to know about their seropositive status at a very late age. The age of disclosure reported in the study was higher than the mean age of 12 years reported in other studies in South Africa and elsewhere [5,19,31,42]. The participants in this study, like those in other studies, stated that the timing of the disclosure to them was too late [26,42] and had impacted negatively on their acceptance of their HIV status, because knowing about their HIV status at a younger age would have facilitated their acceptance of their HIV status.

Consistent with prior findings, the study established that the reactions and responses to disclosure are shaped by the social context under which disclosure of HIV status occurs [14,27]. In this study, disclosure to the participants was delayed, was a one-time event, was unplanned, the participants were not prepared, and they were not ready for the disclosure. Similar findings were reported in other previous studies [5,16,42]. Disclosure was commonly triggered by the severity of the ill-health experienced, which explains the feelings of the fear of death and dying reported by the participants. Similar observations have been reported elsewhere [5,16]. The findings highlight the need to skill HCWs and caregivers to adopt a process-oriented approach to disclosure and foster post-disclosure psychosocial support [16].

The participants experienced disclosure as a traumatic event, and their responses were emotionally charged. They described immediate feelings of distress, sadness, pain, anger, fear, confusion, self-blame, disbelief, and shock post-disclosure. Other studies reported similar negative emotions in the immediate post-disclosure period [5,16,19,31,42,43]. The study further established that the feelings of anger, pain, and sadness persisted long after disclosure occurred. Adolescents in a recent Kenyan study revealed that the process of self-acceptance was prolonged, and the negative emotions took more than a year to resolve [31]. Additionally, a prolonged period of negative perceptions of the self, denial, internalized stigma, uncertainty about living a normal life, and self-isolation was observed among the participants in the current study. Other researchers reported similar observations [20,44].

The participants in this study, blamed their biological mothers for transmitting HIV to them, as well as for delaying the disclosure event. They reported feelings of anger and disappointment, and failed to understand why their caregivers had kept silent and had lied about their diagnosis [13,31,42,43]. The prolonged anger towards the mother could be attributed to the mistrust of parental figures when adolescents feel that deception occurred in the period before disclosure [9]. The current study and others found that the anger towards the mother could not be resolved by the participants whose mothers died before disclosing.

Concerning self-disclosure to others, for most of the participants, acceptance of their HIV status did not lead to self-disclosure of their status to others. Researchers suggest that adolescents with PHIV do not fully accept living with HIV. They alternate between periods of acceptance and moments of distress, and not being able to adapt [45,46]. Those who make pronouncements that they had accepted their HIV status were uncertain about living a normal life. They lived in constant fear of having their serostatus being found out by their peers at school, extended family members, and romantic partners [7,45,47]. The participants developed a sense of fear of self-disclosure for fear of prejudice, stigma, rejection, exclusion from social spaces, and loneliness. The decision not to self-disclose reinforces the secrecy around the HIV diagnosis [45,48].

Researchers agree that self-disclosure to sexual partners is extremely challenging for adolescents with PHIV [5,7,45,48,49]. In late adolescence, the need to be accepted by peers increases, and being accepted as normal is very important [5,19]. The current study suggests that the social belief that HIV is like any other disease might have influenced the participants to say they have accepted their HIV status and are living a normal life. The decision not to self-disclose was taken despite the participants’ belief that people living with HIV are treated well in communities.

Consistent with the findings of other studies, the participants were aware that treatment adherence contributes to the maintenance of their health and helps them to avoid death [5,29,45,50]. They admitted that ART contributes to a normalization of HIV and makes it possible to live a close-to-normal life. However, this study and others found that negative emotions and poor perceptions of the self, undermine treatment adherence [7,9,26,51,52].

The participants described taking ART as a painful, constant reminder that they are living with HIV. They disliked the regimen’s routine and schedules that interfere with their lives, force them to lie to their friends, and take away their ability to enjoy life. Research indicates that post-disclosure, adolescents begin to associate the treatment with being HIV-positive. Hence, the decline in adherence [26]. Given that the fear of self-disclosure is instituted in the daily lives of adolescents with PHIV [48], they take their medication in secret to try to fit into society [26], which further impacts their adherence.

## 5. Limitations of the Study

One of the key limitations of the study is that social desirability cannot be completely ruled out from the responses, since the adolescents might have provided explanations that reflected what they thought the researchers wanted to know. For example, they could have over-exaggerated their acceptance of their HIV status to fit in with the expectations of society. Secondly, the sample was small and came from two health facilities offering ART in the district, despite the small sample size, the study findings have shed light on the long term impact of disclosure on adolescents. In addition, some of the adolescents became upset when they recalled their experiences of the disclosure of their HIV diagnosis, which led to the researchers having to cut their interviews short and refer the adolescents for psychological support that was provided by the resident social worker in all the facilities.

## 6. Conclusions

Despite the persistence of negative emotional outcomes observed among some of the participants following disclosure, others described the disclosure as a positive event. They indicated that they had accepted their HIV status and lived similar to other adolescents, that their adherence to ART had improved post-disclosure, and that they planned to self-disclose to their romantic partners and trusted friends in the near future.

The prolonged period of negative emotional reactions post-disclosure underscores the importance of developing appropriate disclosure interventions and fostering ongoing psychosocial post-disclosure support to improve the emotional health of adolescents with PHIV. In addition, there is a need to strengthen the joint involvement of HCWs and caregivers in the disclosure process by providing appropriate resources and support to caregivers and children to mitigate any potential negative effects of disclosure. These strategies should incorporate culturally appropriate self-disclosure interventions for adolescents with PHIV in primary health settings in SSA.

It is equally important to address treatment adherence pre- and post-disclosure for older adolescents. Therefore, disclosure interventions for children and adolescents with PHIV should address the contextual factors undermining adherence, especially when disclosure was delayed to mitigate the risk of a decrease in adherence after disclosure.

Since there is a significant risk of the transmission of HIV to a sexual partner during adolescence, the reluctance of the adolescents to self-disclose to their sexual partners and the poor adherence to ART have implications for the onward sexual transmission of HIV in this population. It is crucial to encourage self-disclosure in this population, to prevent further HIV transmission.

## Figures and Tables

**Table 1 children-07-00261-t001:** Demographic characteristics of the participants.

Age	Frequency	Percentage
12–14 years	7	33.3
15–17 years	10	47.6
19 years	4	19.1
Schooling status		
Primary school	3	14.2
Secondary school	17	81.
Tertiary institution	1	4.8
Living arrangements		
Living with both parents	5	23.8
Living with mother only	9	42.9
Living with grandparents	6	28.5
Living with siblings	1	4.8

**Table 2 children-07-00261-t002:** Age first tested, age of disclosure, and age when started taking antiretroviral therapies (ARTs).

Variables	Number	Percentage
Age when first tested		
10–14 years	9	42.9
15–19 years	7	33.3
Do not remember	5	23.8
Age of disclosure		
10–14 years	9	42.9
15–19 years	12	57.1
Age when started taking ART		
10–14 years	10	47.6
15–19 years	10	47.6
Do not remember	1	4.8

**Table 3 children-07-00261-t003:** Summary of themes and subthemes.

Theme	Subtheme
Informing adolescents about their HIV-positive status	Circumstances leading to disclosure
	Preferred ideal age of disclosure
	Reaction and response to disclosure
Post disclosure experiences	Self-disclosure of status to others
	Coping with the HIV diagnosis
	Negative perception of self
	Being the only HIV positive sibling
	Taking ART interferes with adolescent’s life

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
