# Peer review of "Pain, Anger, and the Fear of Being Discovered Persist Long after the Disclosure of HIV Serostatus among Adolescents with Perinatal HIV in Rural Communities in South Africa"

_children, 2020, doi:10.3390/children7120261_

Round 1

Reviewer 1 Report

Overall, I thought that the article had merit. I did not see a hypothesis or why this was novel to the other articles published.

Please respond as to the novelty of this study. Could it be that a rural area could be different to an urban area as to how adolescent might respond to the being HIV positive?

Author Response

We thank the reviewer for this comment and we have addressed it to the best of our ability.

Reviewer 2 Report

This manuscript by Maseko Y et.al studied the experience, emotions, and behaviors of adolescents with perinatal HIV when they learned about their seropositive status at a late age. The authors conducted an interview of 21 adolescents (with perinatal HIV) in a rural setup. Based on the interview and questionnaire, they identified mixed emotions of anger, pain, sadness, negative perceptions of self, internalized stigma, and denial persisted even after a long time of disclosure. The idea of this study is great and the manuscript addresses an important topic. However, this manuscript has some significant issues that need to be taken care of. Following are some points

  1. The title of the manuscript is an issue. Please consider redrafting a concise title.
  2. Line no. 96-103 (at the end of the introduction), instead of this sentence, kindly mention the objective of this study. Here please state the main reason for conduction this study. Please redraft the sentences.
  3. The readers will surely fail to understand figure 1. Neither there is figure legend nor explain properly. Kindly reconstruct the figure for a better understanding of the reader. Moreover, there is an overlapping in the subtheme. Why there is a repetition of “negative perception of self”. I think the authors have done a poor job here. Please validate the subtheme.
  4. In the material method section line no. 116-117, “The two facilities were selected because they have a high volume of adolescents with PHIV on ART”. If these facilities have a high number of adolescents, why there are a small number of participants recruited for this study, and why not more samples.
  5. In the result section, unfortunately, although very detailed, this study includes numerous redundancies and repetitions and lacks clarity and proper editing. Some parts just read as a summary of data gathered together, without clear structural and connecting lines. Please consider reorganizing the results in a more streamlined way. There are too many discussions in Results.
  6. Line no 438 “The study found that disclosure had been delayed beyond the recommended age of disclosure to such children”. In my opinion, this study does not fit to conclude such a solid statement. I would advise softening the current notion e.g. participants of this study consider that they came to know about their seropositive status at a very late age.
  7. The authors should have focused more on discussing the delayed disclosure of seropositive status resulted in the mixed emotion of anger, pain, sadness, negative perceptions of self, internalized stigma, etc.
  8. There are numerous repetitions of statements in the discussion.
  9. In the limitation section, please consider including the small number of participants as another limitation of this study.
  10. Please consider acknowledging the research assistant who conducted the interview, caregivers, the staff of the clinics, participants, etc. who helped in completing this study.

Minor comments

  1. There is numerous typo error in the manuscript. Please consider correcting these issues.
  2. Please remove the full stop in line no. 104.
  3. Please consider removing the track change from line no 211.

Author Response

We thank the reviewer for the valuable comments. Most of the comments are addressed in the relevant sections of the manuscript highlighted in blue colour font and also outlined in this document.   

Round 2

Reviewer 2 Report

I thank the authors for attending to my criticisms. In substance, they have responded and addressed most of the concerns in a satisfactory manner but not all. They have not corrected the numerous typo errors, sentence errors/ calculation mistake in the manuscript. Please consider correcting these issues.

For examples;

  1. In the newly added sentences line no 95, there is the repetition of wording like “those in rural settings” etc.
  2. Line no 118-120, the newly added line “The two facilities were selected because they have a high volume of adolescents with PHIV on ART, those who had been informed and those who had not been informed about their HIV diagnosis” does not carry any meaning.
  3. Table no 1 as well as Table no 2. All the percentage calculations are wrong. Kindly recheck these sections thoroughly. Please consider correcting. Please be careful and very precise while reporting something.
  4. This manuscript needs thorough sentence checking and editing. please consider correcting. 

Author Response

Author response 

In the newly added sentences line no 95, there is the repetition of wording like “those in rural settings” etc.

Response: We deleted the repeated word “those in rural settings

Line no 118-120, the newly added line “The two facilities were selected because they have a high volume of adolescents with PHIV on ART, those who had been informed and those who had not been informed about their HIV diagnosis” does not carry any meaning.

Response: sentence corrected-we deleted “those who had been informed and those who had not been informed about their HIV diagnosis”.

Table no 1 as well as Table no 2. All the percentage calculations are wrong. Kindly recheck these sections thoroughly. Please consider correcting. Please be careful and very precise while reporting something.

Response: we checked the calculations and made sure that they add up to 100%.

This manuscript needs thorough sentence checking and editing. please consider correcting. 

Response: we proof read the manuscript.